# CLIPPING FREE ATTACKS AGAINST NEURAL NETWORKS

## ABSTRACT

During the last years, a remarkable breakthrough has been made in AI domain thanks to artificial deep neural networks that achieved a great success in many machine learning tasks in computer vision, natural language processing, speech recognition, malware detection and so on. However, they are highly vulnerable to easily crafted adversarial examples. Many investigations have pointed out this fact and different approaches have been proposed to generate attacks while adding a limited perturbation to the original data. The most robust known method so far is the so called C&W attack [1]. Nonetheless, a countermeasure known as feature squeezing coupled with ensemble defense showed that most of these attacks can be destroyed [6]. In this paper, we present a new method we call Centered Initial Attack (CIA) whose advantage is twofold : first, it insures by construction the maximum perturbation to be smaller than a threshold fixed beforehand, without the clipping process that degrades the quality of attacks. Second, it is robust against recently introduced defenses such as feature squeezing, JPEG encoding and even against a voting ensemble of defenses. While its application is not limited to images, we illustrate this using five of the current best classifiers on ImageNet dataset among which two are adversarialy retrained on purpose to be robust against attacks. With a fixed maximum perturbation of only 1.5% on any pixel, around 80% of attacks (targeted) fool the voting ensemble defense and nearly 100% when the perturbation is only 6%. While this shows how it is difficult to defend against CIA attacks, the last section of the paper gives some guidelines to limit their impact.

## 1   INTRODUCTION

Since the skyrocketing of data volumes and parallel computation capacities with GPUs during the last years, deep neural networks (DNN) have become the most effective approaches in solving many machine learning problems in several domains like computer vision, speech recognition, games playing etc. They are even intended to be used in critical systems like autonomous vehicle [20], [21]. However, DNN as they are currently built and trained using gradient based methods, are very vulnerable to attacks a.k.a. adversarial examples [2]. These examples aim to fool a classifier to make it predict the class of an input as another one, different from the real class, after bringing only a very limited perturbation to this input. This can obviously be very dangerous when it comes to systems where human life is in stake like in self driven vehicles. Companies IT networks and plants are also vulnerable if DNN based intrusion detection systems were to be deployed [23].

Many approaches have been proposed to craft adversarial examples since the publication by Szegedy et al. of the first paper pointing out DNN vulnerability issue [5]. In their work, they generated adversarial examples using box-constrained L-BFGS. Later in [2], a fast gradient sign method (FGSM) that uses gradients of a loss function to determine in which direction the pixels intensity should be changed is presented. It is designed to be fast not optimize the loss function. Kurakin et al. introduced in [14] a straightforward simple improvement of this method where instead of taking a single step of size  in the direction of the gradient-sign, multiple smaller steps  are taken, and the result is clipped in the end. Papernot et al. introduced in [4] an attack, optimized under L0 distance, known as the Jacobian-based Saliency Map Attack (JSMA). Another simple attack known as Deepfool is provided in[34]. It is an untargeted attack technique optimized for the L2 distance metric. It is efficient and produces closer adversarial examples than the L-BFGS approach discussed earlier. Evolutionary

algorithms are also used by authors in [15] to find adversarial example while maintaining the attack close to the initial data. More recently, Carlini and Wagner introduced in [1] the most robust attack known to date as pointed out in [6]. They consider different optimization functions and several metrics for the maximum perturbation. Their L2-attack defeated the most powerful defense known as distillation [8]. However, authors in [7] showed that feature squeezing managed to destroy most of the C&W attacks. Many other defenses have been published, like adversarial training [4], gradient masking [10], defenses based on uncertainty using dropout [11] as done with Bayesian networks, based on statistics [12], [13], or principal components [25], [26]. Later, while we were carrying out our investigation, paper [19] showed that not less than ten defense approaches, among which are the previously enumerated defenses, can be defeated by C&W attacks. It also pointed out that feature squeezing also can be defeated but no thorough investigation actually was presented. Another possible defense but not investigated is based on JPEG encoding when dealing with images. It has never been explicitly attacked even after it is shown in [14] that most attacks are countered by this defense. Also, to our knowledge, no investigation has been conducted when dealing with ensemble defenses. Actually, attacks transferability between models that is well investigated and demonstrated in [22] in the presence of an oracle (requesting defense to get labels to train a substitute model) is not guaranteed at all when it is absent. Finally, when the maximum perturbation added to the original data is strictly limited, clipping is needed at the end of training (adversarial crafting) even if C&W attacks are used. The quality of crafted attacks is therefore degraded as the brought perturbation during the training is brutally clipped. We tackle all these points in our work while introducing a new attack we call Centered Initial Attack (CIA). This approach considers the perturbation limits by construction and consequently no alteration is done on the CIA resulting examples.

To make it clearer for the reader, an example is given below to illustrate the clipping issue. Figure 1 shows a comparison between CIA and C&W L2 attack before and after clipping on an example, a guitar targeted as a potpie with max perturbation equal to 4.0 (around 1.5%). The same number of iterations (20) is considered for both methods.

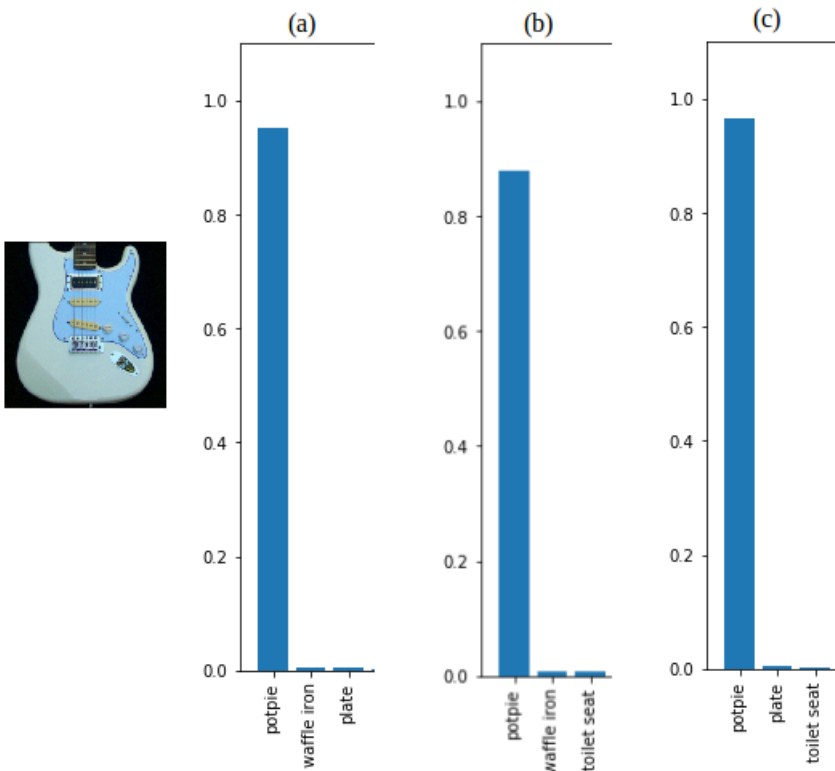

Figure 1: Guitar image targeted as a potpie a) C&W adversarial example classification before clipping, b) C&W adversarial example classification after clipping, c) CIA adversarial example classification.

As can be seen on Figure 1, CIA generates the best attack with 96% confidence. C&W is almost as good with a score of 95% but it is degraded to 88% after applying the clipping to respect the imposed max perturbation. Avoiding this degradation due to clipping is the core motivation of our investigation.

The remaining of this paper is organized as follows. Section I presents some mathematical formulations and the principle of CIA strategy. Then Section II investigates the application of CIA against ensemble defense, feature squeezing and JPEG encoding defenses. Then Section III provides some guidelines to limit the impact of CIA attacks. Finally, we give some possible future investigations in the conclusion.

## 2 Centered initial attack principle

Before entering into details of CIA, let us give some useful formulations.

A neural network can be seen as a function $F(x) = y$ that accepts an input $x$ and produces an output $y$. The function $F$ depends actually on some model parameters often called weights an biases. These are the variables that are adjusted during the learning process to fit the training data on one hand and generalize well to unseen data on the other hand. Since they do not change in our models, we omit them in our notations. The input $x$ can be a vector or an array of any dimension. So, without loss of generality, we consider $x \in \Re^n$ as it can be flattened in any case. So, the $i^{th}$ component of $x$ is noted $x_i$ with integer $i \in [1, n]$.

Since we consider m-class classifiers, the output is calculated using the $softmax$ function. The output $y = F(x)$ can be seen as a vector of m probabilities $p_j$ with $j \in [1, m]$. The component with the biggest value gives the predicted class $C(x)$. This can be written as :

$$C(x) = \arg \max_{j \in [1,m]} \{p_j\}$$

We note the output corresponding to the correct class as $C_c(x)$. An adversarial example $\hat{x}$ is crafted as a non targeted attack so as to get $C(\hat{x}) \neq C_c(x)$ or a targeted one to get $C(\hat{x}) = t$ where $t$ is the target class. Crafting an example can then be formulated using a loss function $L(F, x)$ to maximize the probability of getting a class different from the correct one. Cross entropy is used in our work. The adversarial example $\hat{x}$ can be written as $\hat{x} = x + \delta$ where $\delta$ is the added perturbation. In our work we constrain $\delta$ to be within domain $[-\Delta, \Delta]$ as considered for instance in Google Brain-Kaggle competition [16], $\Delta$ being the maximum perturbation. With the existing approaches, adversarial examples are generated through some iterations then clipped in the end to respect this constraint. With C&W attacks for instance, the loss function includes a norm term $\|\delta\|$ to minimize perturbation $\delta$ but the clipping is still needed as we saw above.

The main idea behind Centered Initial Attack is to find for each component $i$ the center $x_i^*$ of the domain in which $\hat{x}_i$ is allowed to be (green segment on Figure. 2). This is not trivial since we have to insure at the same time the component $\hat{x}_i$ to be within another domain $[\alpha_i, \beta_i]$ to be valid, $\alpha_i$ and $\beta_i$ are respectively the minimum and maximum values that can be taken by the $i^{th}$ feature variable i.e $i^{th}$ component of $x$ and $\hat{x}$. For instance, all components must be in domain [0, 1] for all images in our experiments. Such constraint is needed almost all the time as data are normalized to [0, 1] or [-1, 1] before feeding neural networks.

To sum up, there are two constraints to consider while crafting $\hat{x}$:

a) Domain of definition constraint : $\alpha < \hat{x} < \beta$

b) Max perturbation constraint : $(x - \Delta) < \hat{x} < (x + \Delta)$

To find the center of domain definition of $\hat{x}$, three cases are to be considered actually, not four since $\Delta_i$ is much smaller than $(\beta_i - \alpha_i)$, as can be seen on Figure 2. For recall, $\Delta_i$ is the $i^{th}$ component of $\Delta$.

The three cases are:

**Case (a):** if $(x_i - \Delta_i) > \alpha_i$ and $(x_i + \Delta_i) < \beta_i$ then:

$x_i^* = x_i$

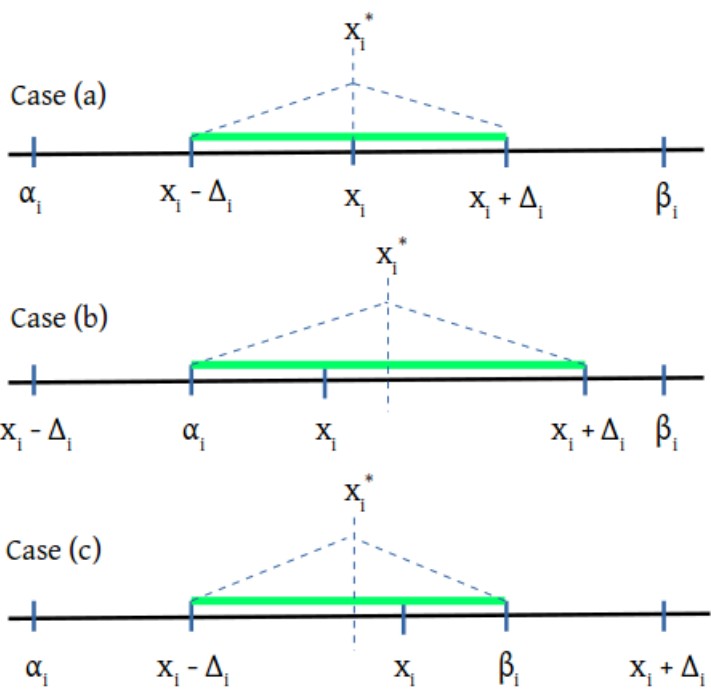

Figure 2: Centered Initial attack principle (thick line is the domain definition of $\hat{x}$).

$\Delta_i^* = \Delta_i$

**Case (b):** if $(x_i - \Delta_i) < \alpha_i$ and $(x_i + \Delta_i) < \beta_i$ then:

$x_i^* = (x_i + \Delta_i + \alpha_i)/2$

$\Delta_i^* = (x_i + \Delta_i + \alpha_i)/2$

**Case (c):** if $(x_i - \Delta_i) > \alpha_i$ and $(x_i + \Delta_i) > \beta_i$ then:

$x_i^* = (x_i - \Delta_i + \beta_i)/2$

$\Delta_i^* = (-x_i + \Delta_i + \beta_i)/2$

Now if we consider a continuous differentiable function $g$ such that: $g : \Re \to [-1, +1]$, then we can write every component $\hat{x}_i$ as:

$$\hat{x}_i = x_i^* + \Delta_i^* \cdot g(r_i) \tag{1}$$

This equation can be rewritten using arrays as:

$$\hat{x} = x^* + \Delta^* \odot g(r) \tag{2}$$

where operator $\odot$ is the elementwise product and $r$ is a new variable on which we optimize the loss function. Finally, the loss function can be written as:

$$L_r = L(F, x^* + \Delta^* \odot g(r)) \tag{3}$$

No constraint is to be considered on the variable $r$ since it is well defined in domain $(-\infty, +\infty)$. Any initialization of $r$ is possible but we consider it as zero for simplicity. So, the initial attack $\hat{x}$ is therefore different from $x$ whereas it is centered in its domain of definition (green segment). This explains the CIA attack name.

Regarding $g$, many continuous functions can be used. For instance we tried three functions:

$g_1(r) = tanh(r)$,

$g_2(r) = 2 \cdot [sigmoid(r) - 0.5]$,

$g_3(r) = sin(r)$.

Obviously other functions can be considered. In our experiments, as they all lead to similar results, we always considered $tanh$.

**Remarks :**

It is interesting to note that with CIA, we can define a different maximum perturbation from a component $x_i$ to another $x_j$. Likewise, it is easy for instance to limit the crafting to only a portion of an image by considering a zero max perturbation on the other regions, without changing anything in the training algorithm. This is an advantage with regard to existing approaches as it is difficult with the current machine learning frameworks to select from the same array only some variables to optimize on. Gradients masking would be a solution but not desired as it is a clipping operation. An example of such partial crafting is displayed on Figure. 3 where only a 50px band on the top and right sides is modified. We generated it using $\Delta = 32$ to make the difference visible on the paper but the image (spider) is also classified as the target (dog) even with smaller values.

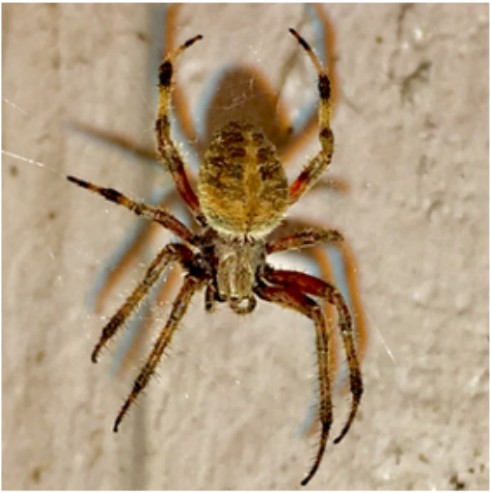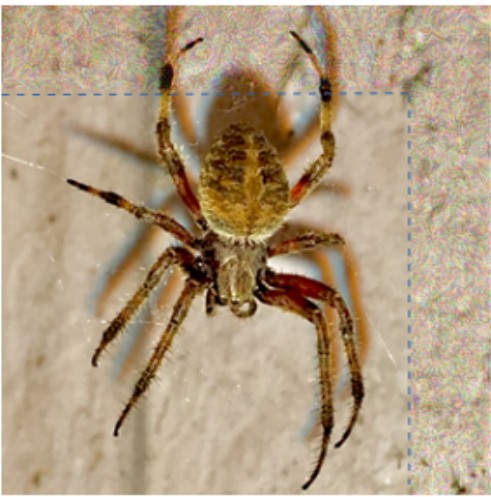

Figure 3: Partially crafted adversarial example(on the right) classified as a 100% dog .

Also, any gradient descent optimizer can be used to craft attacks as the case with [1]. Adam [18] turns out to be the fastest in our experiments. We used it for training and considered 20 iterations, a good compromise between computation time and attacks crafting convergence, for all the adversarial examples crafting. To reproduce the results, the Adam hyperparameters to be considered are $\{learning\ rate = 0.2, \beta_1 = 0.1, \beta_2 = 0.6\}$.

Finally, it is worth noting that CIA is not limited to images. It can be used for any type of data with bounded continuous features.

## 3 APPLICATION

In order to check the effectiveness of CIA attacks, we consider mainly targeted attacks as they are more difficult to craft against three different strategies of defense; ensemble defense with many classifiers, feature squeezing, and JPEG encoding. A combination of these defenses is also considered as we will see later. In this paper, we consider only white box attacks where we have full access to defense models parameters. Other works [22] pointed out the transferability property between models when it is possible to quest the defense classifier and get the labels back to train a substitute model to be used for crafting attacks. When it is the case, an attack generated using the substitute is likely to remain an attack on the defense. When it is not the case however, this transferability is inexistent as we will see below. So, attacking as many models as possible at once is required.

### 3.1 ATTACKING ENSEMBLE DEFENSES

In order to check the robustness of CIA in attacking many classifiers at once, we consider the five best classifiers on ImageNet dataset : Inception V3 (IncV3_a), Inception V4 (IncV4), Inception-Resnet V2 (IncRes_a), adversarialy trained Inception V3(IncV3_b) and adversarialy Inception-Resnet V2 (IncRes_b). The accuracy of these classifiers is around 80% on the whole ImageNet dataset. The accuracies on 1000 images dataset [17] we consider are showed in Table 1.

Table 1: Classifiers accuracies on 1000 images before attacks

| IncV3_a | IncV4 | IncRes_a | IncV3_b | IncRes_b |
|---------|-------|----------|---------|----------|
| 96,1%   | 97,3% | 99,7%    | 94,8%   | 97,5%    |

The first experiment is to attack only one classifier and check the transferability to other classifiers. The attacks are targeted and three maximum perturbation values are considered: 4.0, 8.0, and 16.0. For recall, the images are encoded with unsigned integer values in domain [0, 255]. So, all the results we present take this quantization into account. We first generate the adversarial examples then save them as png files. Finally, we reload them for evaluation.

In this experiment, we attack IncV3_b classifier and present the success rate of targeted attacks and the miss-classification rate of each classifier. The results are showed in Table 2.

Table 2: Results of attacking IncV3_b alone

|  |  | IncV3_a | IncV4 | IncRes_a | IncV3_b | IncRes_b |
|---|---|---------|-------|----------|---------|----------|
| $\Delta = 4.0$ | attacks success rate | 0.0% | 0.0% | 0.0% | 97.6% | 0.0% |
|  | misclassification rate | 4.6% | 3.5% | 0.3% | 98.8% | 3.1% |
| $\Delta = 8.0$ | attacks success rate | 0.0% | 0.0% | 0.0% | 98.9% | 0.0% |
|  | misclassification rate | 5.1% | 3.6% | 0.3% | 99.7% | 3.9% |
| $\Delta = 16.0$ | attacks success rate | 0.0% | 0.0% | 0.0% | 99.1% | 0.0% |
|  | misclassification rate | 7.0% | 4.4% | 0.6% | 99.9% | 5.5% |

As can be seen on Table 2, the transferability is inexistent whatever the maximum perturbation used when looking at the targeted attacks success rate. However, a small increase in misclassification rate is noticed especially with IncV3_a, raising from 3.9% to 7%. This was verified when attacking any other classifier alone and checking the impact on the others. This demonstrates clearly the need to attack many classifiers at once.

To do so, we considered an optimization using a sum of losses, each loss being related to one classifier:

$$L_{ens} = \sum_i L(F_i, x^* + \Delta^* \odot g(r)) \tag{4}$$

where $F_i$ is the function relative to the $i^{th}$ classifier. A weighed version can be considered to target a classifier more aggressively than another. In our experiments, they are all attacked equally. The results are displayed in Table 3.

Table 3: Results of attacking all five classifiers at once

|  |  | IncV3_a | IncV4 | IncRes_a | IncV3_b | IncRes_b | Vote |
|---|---|---------|-------|----------|---------|----------|------|
| $\Delta = 4.0$ | attacks success rate | 95.6% | 59.2% | 30.6% | 92.5% | 32.2% | 83.0% |
|  | misclassification rate | 97.2% | 76.0% | 57.0% | 96.2% | 59.4% | 84.2% |
| $\Delta = 8.0$ | attacks success rate | 99.4% | 85.8% | 66.5% | 94.4% | 53.5% | 96.9% |
|  | misclassification rate | 99.5% | 94.6% | 85.4% | 98.0% | 79.9% | 97.3% |
| $\Delta = 16.0$ | attacks success rate | 100% | 96.8% | 92.5% | 95.2% | 66.6% | 99.8% |
|  | misclassification rate | 100% | 98.9% | 98.3% | 99.2% | 86.8% | 99.8% |

As we can see on Table 3, the success rate of the targeted attacks against the voting ensemble is high, around 80% for $\Delta = 4.0$ and approaching 100% for $\Delta = 16.0$. It is also interesting to notice that the success rate of attacking IncV3_b has decreased compared to the case when it was attacked

alone. With $\Delta = 4.0$ for instance, it went from 97.6% to 92,5%. This can be explained by the fact that the gradients are balanced in a way to change the input in a direction that minimizes all the losses at the same time.

As a conclusion of this section, attacking an ensemble defense using CIA is effective when we have complete access to all defense models (white box attacks). Section 3.3 will show that the transferability to an anknown model, while attacking four among the five, is limited but not negligible (more than 30%).

### 3.1.1 ATTACKING FEATURE SQUEEZING DEFENSE

This defense approach has been developed to counter attacks using smoothing filters [7]. The intuition behind this idea was that smoothing removes the sharp changes brought while crafting adversarial examples. There are different feature squeezing possibilities but we consider spatial smoothing in the current study. The other ones will be addressed in future experiments.

While adding a filter, one should care about the possible loss of accuracy of the defense classifier. Authors in [7] showed that a 3x3 filter is a good compromise that gives an effective defense while limiting the loss of accuracy. We noticed it too in the current investigation and therefore considered this kernel size. Also, different smoothing strategies are possible like Gaussian, diagonal, mean, etc. As the results are quite similar in our experiments we carried out, we consider the mean filter in the sequel for its calculation simplicity. The filter used for defense can actually be replaced by a convolution layer before the neural network as shown in Figure. 4.

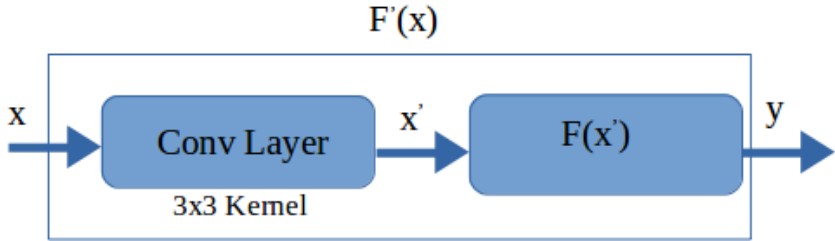

Figure 4: spatial smoothing modeling using a convolution layer.

As can be seen on Figure 4, adding a convolution layer results in a new network that could be represented using a new function $F'$. We can therefore simply craft an adversarial example using this new function. As a start, we conducted an attack against only one network (IncV3_a). The results are shown in Table 4.

Table 4: Results of attacking IncV3_a defending using spatial smoothing

|  |  | IncV3_a | IncV4 | IncRes_a | IncV3_b | IncRes_b |
|---|---|---|---|---|---|---|
| $\Delta = 4.0$ | attacks success rate | 95.4% | 0.0% | 0.0% | 0.0% | 0.0% |
|  | misclassification rate | 99.7% | 6.5% | 2.0% | 10.3% | 6.1% |

As we can notice in Table 4, the success rate of targeted attacks and the miss-classification rate are nearly 100%. This means that the spatial smoothing based defense is not effective. Once again, the transferability between models is inexistent.

An interesting point is to check the success of the same attacks when the defense is not actually using spatial smoothing for sure. Said in other words, we suspect the defense to use spatial smoothing but we are not sure about it. So, we craft attacks as before for a guarantee. Or not! The results are showed in Table 5.

As we can clearly see, the attacks success rate depleted to only 3.2% and the misclassification rate to 18.2%. The attack is therefore not effective in case of filter based defense uncertainty. This result disagrees with the intuition behind spatial smoothing as an efficient defense against attacks presented in [7]. As demonstrated indeed, the filtered adversarial example can be an effective attack but not the unfiltered one !

Table 5: Results of attacking InceV3_a with defense not using spatial smoothing

|  |  | IncV3_a | IncV4 | IncRes_a | IncV3_b | IncRes_b |
|---|---|---|---|---|---|---|
| $\Delta = 4.0$ | attacks success rate | 3.2% | 0.0% | 0.0% | 0.0% | 0.0% |
|  | misclassification rate | 18.2% | 3.6% | 0.5% | 6.4% | 3.3% |

Then, how to overcome this issue for a more robust attack ?

To answer this question, we consider a hybrid network where both filtered and non filtered inputs are used for optimization as represented on Figure. 5.

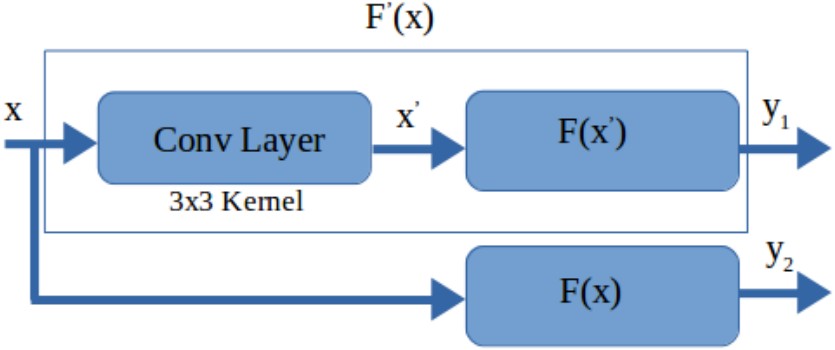

Figure 5: Hybrid network for filter based uncertain defense.

The loss function to be used is given as a sum of two terms (weighing would be useful for more robustness) as follows:

$$L_H = a * L(F, x^* + \Delta^* \odot g(r)) + b * L(F', x^* + \Delta^* \odot g(r)) \tag{5}$$

where a, b are real positive numbers.

We conducted a new experiment using this hybrid loss function and the results are given in Table 6.

Table 6: Results of attacking IncV3_a with a hybrid loss function

| No Filter in defense |  | IncV3_a | IncV4 | IncRes_a | IncV3_b | IncRes_b |
|---|---|---|---|---|---|---|
| $\Delta = 4.0$ | attacks success rate | 99.9% | 0.0% | 0.0% | 0.0% | 0.0% |
|  | misclassification rate | 99.9% | 3.8% | 0.3% | 5.9% | 2.9% |
| 3x3 Filter used in defense |  | IncV3_a | IncV4 | IncRes_a | IncV3_b | IncRes_b |
| $\Delta = 4.0$ | attacks success rate | 98.4% | 0.0% | 0.0% | 0.0% | 0.0% |
|  | misclassification rate | 98.7% | 6.6% | 2.7% | 9.4% | 6.3% |

Table 6 shows that the success rate is this time very high in both cases: 98.4% in filter based defense and nearly 100% in no filter defense. We conclude that this attack is robust whether filtering is used or not in defense.

Another question arises from previous results given the lack of transferability of attacks between models. What if an ensemble defense is used and filter use is uncertain ?

Once again, we consider the sum of all losses, but use the hybrid losses this time as follows:

$$L_{ens} = \sum_i L_{H_i} \tag{6}$$

where $L_{H_i}$ is the hybrid loss relative to the $i^{th}$ classifier.

The results are presented in Table 7

Table 7: Results of attacking all five classifiers with a hybrid losses

| No Filter in defense | | IncV3_a | IncV4 | IncRes_a | IncV3_b | IncRes_b | Vote |
|---|---|---|---|---|---|---|---|
| $\Delta = 4.0$ | attacks success rate | 93.7% | 59.0% | 34.9% | 89.2% | 29.4% | 78.9% |
| | misclassification rate | 96.2% | 76.5% | 58.9% | 94.7% | 54.6% | 81.8% |
| 3x3 Filter used in defense | | IncV3_a | IncV4 | IncRes_a | IncV3_b | IncRes_b | Vote |
| $\Delta = 4.0$ | attacks success rate | 78.0% | 43.1% | 26.3% | 30.6% | 10.3% | 43.1% |
| | misclassification rate | 87.7% | 67.4% | 51.4% | 52.5% | 30.6% | 52.9% |

Even with $\Delta = 4.0$ and considering only targeted attacks, the success and miss-classification rates are high, around 50% when all classifiers use filters and much higher (around 80%) when no filters are used. Other experiments we conducted showed that attacks rate for filter based defense can be improved by assigning a greater weight $b$ (twice the weight $a$) in the hybrid loss equation (5).

## 3.2 ATTACKING JPEG ENCODING DEFENSE

An investigation conducted by authors in [14] showed that most adversarial examples are countered if they are JPEG encoded before classifying them. Lets check if it is the case with CIA attacks. We conducted an attack against IncV3_a and classified the adversarial examples after being JPEG encoded. The encoding uses different compression quality values $Q$. A higher $Q$ means a better quality of image with a bigger size however since it undergoes less loss and compression. The results are displayed on Table 8.

Table 8: Results of attacking JPEG encoding defense

| $\Delta = 16.0$ | | Q = 80 | Q = 70 | Q = 50 | Q = 20 |
|---|---|---|---|---|---|
| Non targeted attacks | | 99.3% | 96.6% | 84.8% | 43.4% |
| Targeted attacks | Success rate | 20.4% | 3.6% | 0.2% | 0.0% |
| | Mis-classification rate | 60.5% | 44.7% | 33.3% | 24.1% |

Table 8 shows that CIA is robust when performing non targeted attacks as almost 100% of them are successful with $Q = 80$ and around 50% with $Q = 20$. The targeted attacks are less successful with the highest score of 20% when $Q = 80$ and 0% when $Q = 20$.

The result is somehow mitigated with regard to targeted attaks results. Indeed, one has to keep in mind that a $Q = 20$ would not be a reasonable defense as this will degrade highly the accuracy of the classifier. Nonetheless, we tried to improve the attacks success score by finding a suitable approximation JPEG transformations.

Obviously, JPEG encoding cannot be modeled accurately using a differentiable function that can be included in crafting examples process as we did before with spatial smoothing. As a brief recall, this encoding implies the passage from RGB space to another color space called YCbCr where Y is brightness, Cb and Cr components represent the chrominance. Actually, humans can see considerably more fine detail in the brightness of an image (the Y component) than in the hue and color saturation of an image (the Cb and Cr components). Considering this fact, Cb and Cr can be downsampled by a factor of 2 or 3 without sensitive change of receptivity to human eye. Another fact is the eye not being sensitive to sharp changes in images. So, removing the high frequencies from the spectral space after a DCT (Discrete Cosine Transform) of an image would not affect its quality remarkably too. These are the important facts used when making a JPEG compression. Other steps like dividing the image into blocks, the quantization of frequency components, the encoding of these components and so on are thoroughly well documented for interested readers [24]. We do not take them into account.

Our idea for approximating JPEG is represented in Figure 6.

As shown on Figure 6, we first transform RGB images to YCrCb space using a function $T$ (product an sum operations) then we filter each component using a convolution layer. Given the facts enumerated before about filtering high frequencies and down sampling the chrominance, we consider a mean 3x3 kernel for brightness component and a 6x6 kernel for Cr and Cb. Once filtered, the result is brought back to RGB space using $T^{-1}$ before feeding the neural network.

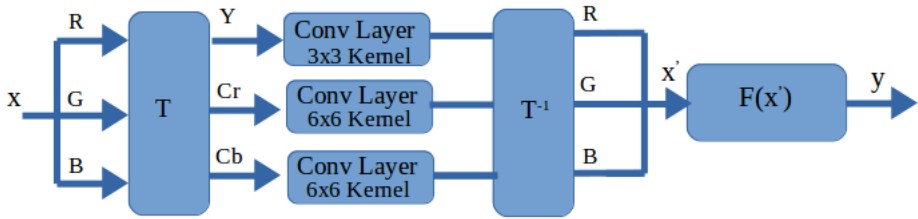

Figure 6: JPEG approximation scheme.

The results of crafting attacks against IncV3_a using this approximation are given in Table 9.

Table 9: Results of attacking JPEG encoding defense using approximation

| $\Delta = 16.0$ | | Q = 80 | Q = 70 | Q = 50 | Q = 20 |
|---|---|---|---|---|---|
| Non targeted attacks | | 99.2% | 97.2% | 83.5% | 44.9% |
| Targeted attacks | Success rate | 20.5% | 3.0% | 0.3% | 0.0% |
| | Mis-classification rate | 59.9% | 44.3% | 34.8% | 23.2% |

As can be noticed, the results are almost the same as those of Table 8. This result is a bit untriguing as if all the filters used in the approximation are inexistent. However, a similar remark as in the filter-based defense case can be made. The attacks crafted using JPEG encoding are no longer attacks if JPEG is not used by the defense. This means that the considered approximation of JPEG encoding is not that bad but not enough accurate to give strong attacks. It has obviously to be improved. We are working on it.

## 3.3 DEFENSE AGAINST ATTACKS

As we saw previously, defending against attacks is hard as along as the defense can be modeled using a function that could be added to form a new network that can be used for crafting examples. As a conclusion, one has to find a transformation that is hardly represented or approximated using simple functions. This is obviously not trivial as we saw with JPEG defense.

Another, and more short term realizable defense, is to consider a big number of well performing classifiers including limited accuracy variance transformations like spatial smoothing. As we saw in Table 7, the success rate decreased to around 50% when using five classifiers including filters for defense. This is not guaranteed but it worth being investigated.

For instance, when we attack all the classifiers except IncV4, the transferability is somehow limited. Indeed, the success rate of attacks is almost 100% against the attacked classifiers whereas it is only around 34% for IncV4 as can be noticed on Table 10.

Table 10: Results of attacking all classifiers but IncV4

| $\Delta = 16.0$ | IncV3_a | IncV4 | IncRes_a | IncV3_b | IncRes_b | Vote |
|---|---|---|---|---|---|---|
| attacks success rate | 99.9% | 34.0% | 98.3% | 99.9% | 98.2% | 96.9% |

## 4 CONCLUSION

In this paper we presented a new strategy called CIA for crafting adversarial examples while insuring the maximum perturbation added to the original data to be smaller than a fixed threshold. We demonstrated also its robustness against some defenses, feature squeezing, ensemble defenses and even JPEG encoding. For future work, it would be interesting to investigate the transferability of CIA attacks to the physical world as it is shown in [14] that only a very limited amount of FGDM attacks, around 20%, survive this transfer. Another interesting perspective is to consider partial crafting attacks while selecting regions taking into account the content of the data. With regard to images for instance, this would be interesting to hide attacks with big but imperceptible perturbations.

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
