# OpenReview forum: "Clipping Free Attacks Against Neural Networks"
_ICLR.cc/2018/Conference — Reject_

### Official Review · AnonReviewer3 · 2017-11-27
**Clipping Free Attacks Against Neural Networks**

**Rating:** 3
**Confidence:** 3

**Review:**

The paper is not anonymized. In page 2, the first line, the authors revealed [15] is a self-citation and [15] is not anonumized in the reference list.

---

> ### Author Response · Authors · 2017-12-19
> **Our response to AnonReviewer3**
>
> We honestly do not understand your evaluation since sentence “Evolutionary algorithms are also used by authors in [15] to find adversarial examples ...” is not a self citation. Believe us that we are not Nguyen et al and we are not related  to them at all. You will see it clearly when the names well be unveiled.
> So, please take the time to reconsider your evaluation and give us a fair review as our paper is the result of several months of work.

---

### Official Review · AnonReviewer2 · 2017-11-28
**Interesting reparametrization, but too little experimental support**

**Rating:** 4
**Confidence:** 3

**Review:**

This paper presents a reparametrization of the perturbation applied to features in adversarial examples based attacks. It tests this attack variation on against Inception-family classifiers on ImageNet. It shows some experimental robustness to JPEG encoding defense.

Specifically about the method: Instead of perturbating a feature x_i by delta_i, as in other attacks, with delta_i in range [-Delta_i, Delta_i], they propose to perturbate x_i^*, which is recentered in the domain of x_i through a heuristic ((x_i ± Delta_i + domain boundary that would be clipped)/2), and have a similar heuristic for computing a Delta_i^*. Instead of perturbating x_i^* directly by delta_i, they compute the perturbed x by x_i^* + Delta_i^* * g(r_i), so they follow the gradient of loss to misclassify w.r.t. r (instead of delta).

+/-:
+ The presentation of the method is clear.
+ ImageNet is a good dataset to benchmark on.
- (!) The (ensemble) white-box attack is effective but the results are not compared to anything else, e.g. it could be compared to (vanilla) FGSM nor C&W.
- The other attack demonstrated is actually a grey-box attack, as 4 out of the 5 classifiers are known, they are attacking the 5th, but in particular all the 5 classifiers are Inception-family models.
- The experimental section is a bit sloppy at times (e.g. enumerating more than what is actually done, starting at 3.1.1.).
- The results on their JPEG approximation scheme seem too explorative (early in their development) to be properly compared.

I think that the paper need some more work, in particular to make more convincing experiments that the benefit lies in CIA (baselines comparison), and that it really is robust across these defenses shown in the paper.

---

> ### Author Response · Authors · 2017-12-19
> **Our response to AnonReviewer2**
>
> Thanks a lot for taking the time to read the paper and provide us with you review.
> We do not claim in the paper that CIA attacks are the most robust ones as we did not indeed give any comparison to other methods. We however show that they are an answer to some issues met in literature. First, avoid the clipping that degrades the quality of attacks. We give a comparison to C&W. (Figure 1).  Second, we show that CIA attacks are effective against recent published defenses : ensembling (by the way, at least three papers submitted to ICLR2018 claim this defense to be effective), smoothing and JPG encoding. After the paper submission, we continued our experiments and made comparison to C&W and FGSM. They show a non negligible improvement in attacks success using CIA approach. If adding the results would change your review to an acceptance, we would like to do it.
> About the grey-box attack, you are definitely right . However, the purpose of this section 3.3 was to show that ensembling can be considered as a defense as it limits the attacks but not totally effective. Using another classifier would likely show that the transferability is even more limited. This would only reinforce our claim about the lack of transferability which was already tackled in the previous sections.
> Finally, the English of the paper can be improved. We will do it in the revised version.

---

### Official Review · AnonReviewer1 · 2017-11-28
**Incremental but interesting results for adversarial examples**

**Rating:** 5
**Confidence:** 2

**Review:**

In this paper the authors present a new method for generating adversarial examples by constraining the perturbations to fall in a bounded region.  Further, experimentally, they demonstrate that learning the perturbations to balance errors against multiple classifiers can overcome many common defenses used against adversarial examples.

Pros:
- Simple, easy to apply technique
- Positive results in a wide variety of settings.

Cons:
- Writing is a bit awkward at points.
- Approach seems fairly incremental.

Overall, the results are interesting but the technique seems relatively incremental.

Details:

"To find the center of domain definition..." paragraph should probably go after the cases are described.  Confusing as to what is being referred to where it currently is written.

Table 1: please use periods not commas (as in Table 2), e.g. 96.1 not 96,1

inexistent --> non-existent

---

> ### Author Response · Authors · 2017-12-19
> **Our response to AnonReviewer1**
>
> First, thank for taking the time to read the paper and provide this review.
> The approach is not really incremental as we do not go from an easy case then harden it at each new experiment. In the Tabl2 we show the non transferabity of attacks when making targeted attacks which is against what is often claimed in literature. Table 3 gives the results of attacking several classifiers at once.  This shows that ensembling is not always effective as often claimed (by the way, at least three papers submitted to ICLR2018 claim it!). Table 4 provides results of attacking another defense, i.e.  spatial smoothing. Table 5 shows that smoothed attacks are not necessarily successful if defense does not use smoothing. Once again, this reveals that the idea behind smoothing as an efficient defense is not true. Table 6 gives the results of attacking a defense with and without smoothing at the same time. Table 7 presents a combination of ensembling and smoothing defense.  We could have given this last table directly at the beginning but we think honestly that this would make the paper more difficult to read.
> The paper is obviously not written in a perfect English and this can be improved. We will do it in the revised version. But overall we think that we bring some interesting results to the community:
> - Avoid clipping to perform more robust attacks
> - Perform effective attacks against strong defenses like ensembling and smoothing.
> - Make partial crafting without affecting the whole content of input data (images in Figure 3)
> - Finally, CIA attacks can be applied beyond images.
> We hope this answer will give you satisfaction and change your review to an acceptance.

---

### Decision · Program_Chairs · 2018-01-29
**ICLR 2018 Conference Acceptance Decision**

**Decision:**

Reject

**Comment:**

The reviewers have various reservations.
While the paper has interesting suggestions, it is slightly incremental and the results are not sufficiently compared to other techniques.
We not that one reviewer revised his opinion